# Comprehensive Approach to Genomic and Immune Profiling: Insights of a Real-World Experience in Gynecological Tumors

**DOI:** 10.3390/diagnostics12081903

**Published:** 2022-08-06

**Authors:** Iván Prieto-Potin, Franklin Idrovo, Ana Suárez-Gauthier, María Díaz-Blázquez, Laura Astilleros-Blanco de Córdova, Cristina Chamizo, Sandra Zazo, Nerea Carvajal, Almudena López-Sánchez, Sandra Pérez-Buira, Carmen Laura Aúz-Alexandre, Rebeca Manso, Jenifer Plaza-Sánchez, Virginia de Lucas-López, Nuria Pérez-González, Sara Martín-Valle, Ion Cristóbal, Victoria Casado, Jesús García-Foncillas, Federico Rojo

**Affiliations:** 1Department of Pathology, CIBERONC, UAM, Fundación Jiménez Díaz University Hospital Health Research Institute, 28040 Madrid, Spain; 2Cancer Unit for Research on Novel Therapeutic Targets, Oncohealth Institute, UAM, Fundación Jiménez Díaz University Hospital Health Research Institute, 28040 Madrid, Spain; 3Translational Oncology Division, Oncohealth Institute, UAM, Fundación Jiménez Díaz University Hospital Health Research Institute, 28040 Madrid, Spain

**Keywords:** next-generation sequencing, PD-L1, TILs, ovary, uterus, cervix, cancer

## Abstract

Gynecological cancer accounts for an elevated incidence worldwide requiring responsiveness regarding its care. The comprehensive genomic approach agrees with the classification of certain tumor types. We evaluated 49 patients with gynecological tumors undergoing high-throughput sequencing to explore whether identifying alterations in cancer-associated genes could characterize concrete histological subtypes. We performed immune examination and analyzed subsequent clinical impact. We found 220 genomic aberrations mostly distributed as single nucleotide variants (SNV, 77%). Only 3% were classified as variants of strong clinical significance in *BRCA1* and *BRCA2* of ovarian high-grade serous (HGSC) and uterine endometrioid carcinoma. *TP53* and *BRCA1* occurred in 72% and 28% of HGSC. Cervical squamous cell carcinoma was entirely HPV-associated and mutations occurred in *PIK3CA* (60%), as well as in uterine serous carcinoma (80%). Alterations were seen in *PTEN* (71%) and *PIK3CA* (60%) of uterine endometrioid carcinoma. Elevated programmed death-ligand 1 (PD-L1) was associated with high TILs. Either PD-L1 augmented in deficient mis-matched repair (MMR) proteins or *POLE* mutated cases when compared to a proficient MMR state. An 18% received genotype-guided therapy and a 4% immunotherapy. The description of tumor subtypes is plausible through high-throughput sequencing by recognizing clinically relevant alterations. Additional concomitant assessment of immune biomarkers identifies candidates for immunotherapy.

## 1. Introduction

Among gynecological cancer, three types account for the vast majority of new cases that are included in the top ten most common female cancers worldwide in 2020: cervix uteri (6.5%), corpus uteri (4.5%) and ovary (3.4%) [1]. Recent data from Spain estimates a 12% rise in new cases in gynecological cancer, thus indicating an elevated incidence and related mortality requiring our attention for an appropriate management and care [2].

Since the implementation of high-throughput sequencing in the clinical setting, several cancer types have improved predictive therapeutic interventions, hence resulting in better prognosis. In some malignancies, such as in lung or in colorectal cancer [3,4], the clinical effects of multiple biomarkers analysis in a single assay have led to the development of numerous targeted treatments. However, in gynecological cancer, the role of next-generation sequencing has brought understanding on its molecular pathogenesis but is still limited regarding its use in the diversity of targeted treatment [5]. In fact, the current use of a comprehensive genomic profiling is revealing relevant clinically alterations that may present diagnostic and prognostic implications, as well as therapeutic options for patients affected by gynecological cancer [6]. A genomic profiling approach now permits a more accurate classification of tumor subtypes than the traditional cancer nomenclature primarily based on cell typing, tumor grading, immune and histopathological marker’s location or identification [7]. For instance, the well-known molecular allocation of endometrial cancer already divides patients on ultramutated, hypermutated, copy number high, and low groups with their respective prognostic and therapeutically clinical consequences regarding tumor subtypes after the TCGA`s integrated genomic analysis [8,9]. This molecular categorization has also been extended to other kind of tumors as occur in epithelial ovarian cancer, precisely regarding the high-grade serous subtype which is segregated in four different assemblages: immunoreactive, differentiated, proliferative, and mesenchymal that correlate with clinical outcome [10,11,12]. Likewise, integrated studies revealed three groups in cervical cancer: keratin-low squamous, keratin-high squamous, and adenocarcinoma-rich clusters defined by HPV and molecular features [13]. Therefore, such a molecular characterization is really influencing gynecological cancer classification as well as having a real impact on a patient’s clinical course.

Current studies are now trying to reproduce genomic signatures in endometrial cancer with more accessible high-throughput sequencing platforms already introduced in the clinical setting. These exhibit lower costs and turnaround time than those technologies employed by the cancer genome atlas (TCGA)’s integrated genomic analysis. However, some studies are not able to replicate the four molecular subtypes by using a customized-NGS panel targeting 156 genes [14]. In fact, others have applied the molecular classification of endometrial cancer to epithelial ovarian cancer with another designed-NGS panel of 47 genes and found genetic heterogeneity being able to classify certain percentage of tumors into the molecular groups [15]. Indeed, more and more studies are now demonstrating the real power of the multiple panels settled in the clinical setting toward a molecular classification of gynecological cancers [16,17]. However, the wide diversity of high-throughput panels does not always allow a matched representation of classical biomarkers in a particular type of gynecological tumor. Hence, selecting appropriate high-throughput sequencing panels toward a precise characterization of gynecological tumor subtypes is crucial to obtaining accurate genomic profiling of a patient’s tumor.

Inflammation is a hallmark of cancer [18]. In fact, the tumor immune microenvironment plays a crucial role in many cancers. The characterization of tumor infiltrating lymphocytes (TILs) in ovarian HGSC may lead to stratify patients according to their prognosis as well as to identify candidates for immunotherapy in case of relapse [19]. Likewise, the analysis of immune checkpoint molecules such as PD-L1 have shown the capability of predicting favorable prognosis in advanced ovarian HGSC [20]. Recent studies demonstrate the importance of characterizing the immune environment of different kinds of gynecological tumor across the different genomic aberrations subtypes to improve response rates and efficacies of applied treatments [21,22]. Therefore, the assessment of immune biomarkers across gynecological cancer would bring finest tailored treatment strategies as well as a more accurate classification of each tumor subtype.

This study aims to determine whether the application of high-throughput sequencing established as part of the clinical care practice is able to characterize particular histological subtype of gynecological tumors by the detection of precise alterations. We also wonder if the use of a comprehensive panel would lead to identify any potential therapeutic target that may benefit the patient’s outcome. In addition to the molecular approach, we extended the tumor characterization through the study of immune markers in order to analyze if the combination of both immune and molecular biomarkers would result more advantageous for the patient outcome as part of the clinical routine.

## 2. Materials and Methods

### 2.1. FFPE Tissue Collection

We compiled data from 49 patients with gynecological tumors (including ovary, uterus, cervix and vagina) who were asked for high-throughput sequencing testing between 2017 and 2019 during a routine molecular diagnostic practice at the Fundación Jiménez Díaz University Hospital. We retrieved archived formalin-fixed, paraffin-embedded (FFPE) tissue material from the Biobank Fundación Jiménez Díaz (PT20/00141) that belongs to the Spanish National Biobank network. Each donor gave written consent. All investigations followed standard operating procedures with the approval of the Fundación Jiménez Díaz University Hospital Ethics and Scientific Committee (PIC209-21) and were performed in accordance to the principles outlined in the Declaration of Helsinki.

### 2.2. DNA and RNA Isolation

FFPE tissues were sectioned 3-µm thick for hematoxylin and eosin staining (Dako coverstainer, Agilent, Santa Clara, CA, USA) to ensure appropriate tumor-cell content. We used consecutive sections to extract both genomic DNA and RNA with Recoverall total nucleic acid isolation kit according to the manufacturer’s instructions (ThermoFisher Scientific, Waltham, MA, USA). We then quantified the purified DNA and RNA with respectively Qubit dsDNA BR assay kit (ThermoFisher Scientific, Waltham, MA, USA) and RNA Screentape Analysis kit (Agilent, Santa Clara, CA, USA).

### 2.3. High-Throughput Sequencing

The multi-biomarker Oncomine Comprehensive Assay (OCA) v3 (ThermoFisher Scientific, Waltham, MA, USA) was used to prepare DNA and RNA libraries in agreement with the manufacturer’s protocol. The panel enabled the detection of relevant single nucleotide variants, copy number variations, gene fusions, and INDELs from 161 cancer-associated genes (Appendix A). Briefly, DNA libraries generation used the Ion Ampliseq library kit 2.0 (ThermoFisher Scientific, Waltham, MA, USA) by incorporating barcode, adapters and RNA libraries employed the Ion Ampliseq RNA library kit (ThermoFisher Scientific, Waltham, MA, USA). Both pooled libraries templates were prepared on the Ion Ampliseq library Ion Chef system with the Ion 540 Chef kit (ThermoFisher Scientific, Waltham, MA, USA). Subsequently, multiplexed libraries templates were sequenced using Ion 540 chips on the Ion Torrent S5 XL platform (ThermoFisher Scientific, Waltham, MA, USA) as indicated by the manufacturer’s protocol. Data analysis for variant calling and annotation were performed with the Ion Reporter software v.5.10. Mapped reads used the hg19 reference genome, a minimum sequencing depth of 500× was considered appropriate and variants with an allelic frequency (AF) less than 5% were filtered and excluded before review, except for certain variants where a 1% AF was contemplated. We considered intergenic fusions with a depth superior than 40× and intragenic fusions with 100×, whereas the limit of copy number variation was set up to more than five copies. Variant categorization was assessed as previously described [23,24]. Briefly, Tier IA and IB categories corresponded to variants with strong clinical significance, Tier IIC and IID to variants with potential clinical significance, and Tier III to variants with unknown clinical significance.

### 2.4. Microsatellite Instability

Detection of microsatellite instability was done through polymerase chain reaction and fragment analysis of a mononucleotide repeat pentaplex panel to identify alterations of repetitive tandem regions as previously described [25]. Briefly, in-house primers used the following sequences for *SLC7A8*, NR21-Fw: 5′-TAAATGTATGTCTCCCCTGG-3′, NR21-Rv: 5′-ATTCCTACTCCGCATTCACA-3′, *ZNF-2*, NR-24-Fw: 5′-CCATTGCTGAATTTTACCTC-3′, NR-24-Rv: 5′-ATTGTGCCATTGCATTCCAA-3′, *inhibitor of apoptosis protein-1* NR-27-Fw: 5′-AACCATGCTTGCAAACCACT-3′, NR-27-Rv: 5′-CGATAATACTAGCAATGACC-3′, *c-KIT* BAT-25-Fw: 5′-TACCAGGTGGCAAAGGGCA-3′, BAT-25-Rv: 5′-TCTGCATTTTAACTATGGCTC-3′ and *MSH2* BAT-26-Fw: 5′-TGACTACTTTTGACTTCAGCC-3′, BAT-26-Rv: 5′-AACCATTCAACATTTTTAACCC-3′) for multiplex amplification of purified DNA. Resulting amplicons were sequenced on the ABI-Prism 3130 DNA analyzer (Applied Biosystems, Foster City, CA, USA). Fragments were analyzed with the Peak Scanner^TM^ 2 software (ThermoFisher Scientific, Waltham, MA, USA) and instability assessment was defined as a loss of stability in two or more repeats out of the five microsatellite markers [26].

### 2.5. HPV Genotyping

Identification of HPV genotypes was executed with the CLART HPV4 kit (Genomica, Madrid, Spain) by following the manufacturer’s protocol. This PCR-based microarray assay targets a preserved region of the *L1* gene and enables the detection of 35 different genotypes of the virus. These are organized in three groups: high risk (16, 18, 31, 33, 35, 39, 45, 51, 52, 56, 58, 59, 66, 68), probably high risk (26, 53, 73, 82), and low risk (6, 11, 40, 42, 43, 44, 54, 61, 62, 70, 71, 72, 81, 83, 84, 85, 89). The Clinical Array Reader and image management software, Saiclart, automatically carried out the analysis and interpretation on the Autoclart platform (Genomica, Madrid, Spain). The assay amplifies human *CFTR* gene and a modified plasmid as internal controls for ensuring validated DNA quality and processing. A second repetition was considered definitive when samples resulted in a first invalid result, either inhibition of PCR or DNA absence.

### 2.6. Immunohistochemistry

Serial FFPE tissue sections of 3 µm thick were employed for immunohistochemical studies in order to determine the protein expression of ERBB2, MLH1, MSH2, MSH6, PD-L1, PMS2, and p53 as depicted in Appendix A. Recommendations of the international TILs working group were followed to evaluate TILs [27,28]. Slides were incubated 1 h at 56 °C and deparaffinized. High pH EnVision FLEX Target Retrieval solution was used for antigen retrieval in the PT-link instrument (Dako, Glostrup, Denmark) for 20 min at 95 °C. Staining with primary antibodies was automatically achieved on the Autostainer Link 48 with the EnVision FLEX System-HRP kit (Dako, Glostrup, Denmark) after endogenous peroxidase activity blocking with the EnVision FLEX peroxidase-blocking reagent for 10 min. Preparations were finally incubated 10 min on chromogen 3,3-diaminobenzidine, automatically counterstained and mounted in the Coverstainer platform (Dako, Glostrup, Denmark). Two different pathologists independently assessed the presence or absence of tumor cell immunoreactivity and positive controls. Evaluation of PD-L1 was done by calculation of the combined positive score (CPS = number of PD-L1 staining cells (tumor cells, lymphocytes, macrophages)/total number of viable tumor cells × 100), as reported earlier [29]. Assessment of p53 was scored by 0-normal or wild-type, 1-abnormal overexpression, 2-abnormal cytoplasmatic expression, and 3-complete absence, as previously described [30]. ERBB2 was graded as 0-no staining, 1+ incomplete membrane staining in <10%, 2+ strong complete membrane staining in ≤30% or incomplete in ≥10%, and 3+ strong staining in >30% of tumor cells [31].

### 2.7. ERBB2 Fluorescent in situ (FISH) Hybridization

We used additional sequential FFPE tissue sections of 3-µm thick to identify *ERBB2* amplification on tumors with a 2+ immunohistochemical score. It also served as the orthogonal approach that confirmed *ERBB2* amplifications resulting from the high-throughput sequencing panel. We performed FISH according to the manufacturer’s instructions (Pathvision, Abbott molecular, Des Plaines, IL, USA), as previously described [32]. We achieved the analysis on a fluorescent microscope DM5500B (Leica Microsystems, Wetzlar, Garmany) using Cytovision software (Leica Microsystems, Wetzlar, Garmany). Assessment was done by counting red signals by nucleus and green signals by nucleus respectively corresponding to *ERBB2* gene and chromosome 17 centromere (CEP17). According to the established criteria, we considered amplification of ERBB2 if the ratio of ERBB2/ CEP17 ≥ 2 or ERBB2/ CEP17 < 2 and the mean signals of ERBB2 by nucleus ≥6 [31].

### 2.8. Statistical Analysis

Statistical analysis used SPSS version 21.0 software for Windows (IBM, New York, NY, USA) and GraphPad Prism version 5.0 software (GraphPad Software, Inc., La Jolla, CA, USA). Descriptive data were expressed as median and interquartile range. Associations were made by using Spearman’s rho rank correlation coefficient and comparisons between multiple groups used Kruskal–Wallis tests with Bonferroni correction of post hoc Mann–Whitney U tests. *p* < 0.05 was considered significant.

## 3. Results

### 3.1. Clinicopathological Characteristics of Cohort

From the 49 female patients included in the study, we identified almost half of them (*n* = 22) with ovarian cancer, *n* = 15 with uterine cancer, *n* = 11 with cervical cancer, and *n* = 1 with vaginal cancer. Only 16% of the patients exhibited metastatic disease at the time of diagnosis and 26% had family history of malignancy. The major histological subtype representation corresponded to high-grade serous ovarian carcinoma (37%), followed by endometrial endometrioid carcinoma (12%) and cervical squamous cell carcinoma (12%). Thirty-five patients (71%) showed advanced-stage disease. Taxane plus platinum treatment was the chemotherapy mostly administered in the cohort (59%) whereas adjuvant radiotherapy was given in 34% of the patients. About 79% underwent surgery debulking and experienced disease recurrence. Additional clinical data displaying radiological response to therapy by tumor site are depicted in Table 1.

### 3.2. Detection of Genomic Alterations across the Cohort

Evaluation of the 49 patients permitted the recognition of 220 variants distributed as single nucleotide variants (SNV, 77%), copy number variation (CNV, 13%), insertion-deletion (INDEL) mutations (7%), gene fusions (2%), and multiple nucleotide variants (MNV, 1%). No variants were detected in 12% of the patients. SNV exhibited a median coverage of 1995× (1333–1999) and a variant allele frequency of 34% (19–51), whereas INDEL mutations and MNV were covered 1973× (1356–1992). CNV alterations showed a median copy number of 7 (6–11) and gene fusions presented a median count of 131 (75–185) reads. Further sequencing raw data information of detected variants can be accessed in Appendix A. Only 3% of detected variants were classified as variants of strong clinical significance (Tier I), 47% were variants of potential clinical significance (Tier II), and 50% were variants of uncertain significance (Tier III). Tier IA variants were found in *BRCA1* and *BRCA2* genes from three ovarian high grade serous carcinomas and one uterine endometrioid adenocarcinoma, respectively (Figure 1A). Tier I and tier II alteration types were more represented in the *TP53*, *PIK3CA*, *ERBB2*, *CCNE1*, *PTEN,* and *STK11* genes (Figure 1B). The number of genomic alterations ranged from 0 to 15 in the whole cohort. Uterine serous and endometrioid carcinomas presented an elevated number of genomic alterations, 7 (4–12) and 6.5 (5–9), respectively, in comparison with ovarian HGSC, 3.5 (3–5), *p* = 0.01 and with cervical HPV-associated carcinoma, 1.5 (0–5), *p* = 0.01 (Figure 1C).

We also compared the frequencies of single somatic mutations-affected cases in the studied cohort against those cases listed in the TCGA database for each tumor site. The *TP53* gene was more frequently altered in the ovary (64% against 91% in TCGA), whereas *PIK3CA* (47% against 51%), *PTEN* (33% against 67%), and *TP53* (27% besides 40%) genes presented more altered variants in the uterus. Cervix exhibited the *PIK3CA* gene as the more altered gene, (33% of the cases against 29% of the TCGA) (Appendix A).

### 3.3. Characterization of Histological Subtypes

We conducted a brief literature review in order to define the molecular biomarkers characterizing precise histological subtypes included in the cohort. Each of them was assigned to a particular signaling pathway. We assessed those cancer-associated genes included in our high-throughput sequencing panel. We were unable to identify mutations in *EMSY*, *BRIP*, *BARD1,* and *FOXM1* CNV in high-grade ovarian carcinoma as well as *PTEN* loss in clear cell carcinoma, *LRPB1* loss and *SOX17* amplification in uterine serous carcinoma (Table 2).

We then analyzed the mutational profiles and protein expression of the histological subtypes. We found that alterations in the *TP53* and *BRCA1* genes occurred in 72% and 28% of the HGSC cases, respectively. Other top altered genes in HGSC included *CCNE1* (17%), *FANCD2* (17%), *POLE* (17%), and *PMS2* (17%). Only one case showed a *PIK3CA* CNV (5%) (Figure 2A). All HGSC cases exhibited abnormal P53 expression except in one case (Figure 3) and were proficient MMR or MSS. Although ovarian clear cell carcinoma was represented in a minor proportion, *PIK3CA* and *ARID1A* genes were found mutated in two of the three studied cases. Only case 403, an ovarian clear cell carcinoma, presented dMMR with a concrete pattern of expression loss in MSH6 (Figure 4). Regarding uterine serous carcinoma, the proportion of cases showing altered genes was *PIK3CA* (80%), *TP53* (40%), *ERBB2* (40%), *FBXW7* (40%) and with less representation *PPP2R1A* (20%) and *FGFR2* (20%). From the two cases showing SNV in *ERBB2*, one revealed concomitant *ERBB2* amplification (20%). Furthermore, as illustrated in Appendix A, additional histological subtypes exhibited *ERBB2* amplification such as ovarian HGSC and cervical adenocarcinoma. Even though, we observed dissimilarities with the orthogonal approach in cases 287 and 393, probably explained by the limit of detection applied in the assessment of the copy number variation from the high-throughput panel. In fact, we just found a 10% of the cohort presenting *ERBB2* SNV and 6% CNV, whereas conventional testing identified up to 8% of *ERBB2* amplification. Concerning endometrial endometrioid carcinoma, cases presenting genes with alterations in *PTEN* (71%), *PIK3CA* (60%), and *NOTCH1* (43%) were the most observed. About 28% of affected cases presented alterations in *CTNNB1*, *KRAS*, *POLE*, *SETD2*, *PIK3R1*, *FANCI,* and *ARID1A* genes (Figure 2B). On the other hand, squamous cell carcinoma cases of the cervix were entirely HPV-associated and mutations occurred in the *PIK3CA* (60%) gene, whereas mucinous carcinomas were HPV-independent and mostly exhibited alterations in the *STK11* gene (Figure 2C and Appendix A).

### 3.4. Immune Characterization of Gynecological Tumors and Detected Altered Genes

Immune microenvironment along the diverse histological subtypes was studied by PD-L1 and TILs presence. We then correlated them with the genotypes obtained with the high-throughput panel. We assessed the number of genomic alterations for each patient with either PD-L1 or TILs scores in order to confirm whether those tumors with an elevated quantity of alterations may present a high score of immune markers. We did not find any association with the PD-L1 score (rho = 0.07, *p* = 0.68), nor with the TILs value (rho = 0.13, *p* = 0.41) (Figure 5A,B). We also evaluated the PD-L1 and TILs in each histological subtypes so we could seek any evidence of an immune phenotype in a particular histology. Ovarian HGSC, uterine endometrioid carcinoma, and cervical HPV-associated cases exhibited the more elevated PD-L1 and TILs scores (Figure 5C–F). An association was found between PD-L1 score and TILs in the entire cohort (rho = 0.48, *p* < 0.01) as depicted in Figure 5G. We finally compared the immune markers of those samples presenting a mismatch repair proficiency with those showing a deficiency state and an alteration in the *POLE* gene. We found elevated PD-L1 scores for dMMR, altered *POLE,* and the sum of both in comparison to the pMMR cases (Figure 5H).

The altered *POLE* cases presented a median TILs score value of 90% (38–94) and the dMMR cases, 75% (15–90). Both kind of cases expressed an elevated score (*p* < 0.01 and *p* < 0.01) in comparison to the pMMR state group which exhibited a 10% (1–60) score (Table 3).

### 3.5. Clinical Impact

We finally studied the utility of clinically relevant alterations and administered targeted therapies to assess the clinical impact of the routine high-throughput sequencing application in gynecological cancer patients (Table 4).

About 18% of the cases received a targeted therapy based on PARP inhibitors (PARPi). Out of the 8% of cases presenting a Tier IA alteration in the *BRCA1* and *BRCA2* genes, only 4% were administered Olaparib treatment resulting in a partial radiological response. Another 10% received PARPi including Olaparib or Niraparib yet only a 6% presented a Tier III alteration in genes of the HR pathway. About 12% of the cases had the PD-L1 score evaluated at the time of diagnosis whereas the rest of the cases were calculated prospectively. From them, only a 4% received pembrolizumab treatment with a successive partial radiological response.

Clinical benefit was considered for those patients achieving complete or partial response whereas non-clinical benefit was reflected as progressive disease. From the five patients presenting *POLE* mutations, three radiological partial responses were observed whereas in two of them it remained lost to follow-up.

### 3.6. Total Cost and Cost Difference versus NGS

We performed an evaluation of the conventional testing strategy versus the high-throughput panel used for those more clinically relevant biomarkers in high-grade serous ovarian carcinoma. The cost per sample, the testing strategy, total cost, and cost difference are depicted in Appendix A. The application of the profiling protocol including the comprehensive NGS panel and the identification of immune biomarkers (PD-L1 and TILs) resulted more profitable than the use of conventional or single gene testing.

## 4. Discussion

In this study, we showed how comprehensive high-throughput sequencing currently established in the clinical care practice enabled the characterization of mutational profiles from histological subtypes of gynecological cancer through the identification of precise alterations in cancer-associated genes. Furthermore, the use of such a comprehensive panel was able to recognize clinically relevant genomic alterations leading to genotype-guided therapy.

In our study cohort, we described histological subtypes with a major representation at the molecular level. Alterations in the *TP53* and *BRCA1* genes occurred in 72% and 28% of the HGSC cases. In fact, it is reported that about 20–30% of HGSC presented a *BRCA* mutation [33]. Standard guidelines identified germline mutations in *BRCA1* and *BRCA2* genes in 15% of women affected with ovarian cancer, whereas somatic mutations in an additional 7% [34]. Instead, a much-elevated proportion is described for *TP53* mutations, ranging from 94% to 97% as a probably observed frequency in previous studies [35]. Our rate in *TP53* mutation (72%) was similar to the observed abnormal protein expression of p53 (78%). We detected 17% of discordant cases with a wild-type genotype exhibiting an aberrant protein expression of p53 that may be explained by the sensitivity of the panel, a mutant allelic frequency less than 5% was excluded for variant review. Considering either *TP53* mutation or p53 abnormal expression analysis, 89% of the HGSC cases were found to be altered, thus complying with the elevated rate widely described in the literature. Another top changed gene in this particular histological subtype contemplated amplifications in the *CCNE1* gene, more observed in cases developing resistance to platinum-based chemotherapy with a frequency of 28% [36]. *CCNE1* amplification in HGSOC is mutually exclusive of homologous recombination pathway mutations, including *BRCA* alterations, resulting in an ineffective PARP inhibition [37]. Our comprehensive analysis detected less copy number variations, only in 17% of the cases. As well, diverse alterations were encountered in genes of the HRR pathway, which is in line with other studies that described alterations in *RAD51C*, *PALB2,* or genes of the Fanconi pathway [38]. In fact, ovarian HGSC and endometrial serous carcinomas share similar molecular alterations [39], nearly each case of endometrial serous carcinoma harbors an alteration in the *TP53* gene or aberrant expression of p53 [40]. However, in our cohort, only 40% of endometrial serous carcinoma exhibited a somatic alteration and 60% an aberrant protein expression. In contrast, an 80% of the cases showed an alteration in the *PIK3CA* gene which is associated with metastasis and is reported in a much lower frequency, between 30% and 60% [41]. Another biomarker characterizing the endometrial serous carcinoma considered amplifications in the *ERBB2* gene in an approximately 25–40% rate and we found between 20 and 40% to be amplified or mutated [42]. Indeed, endometrial serous carcinoma has been found as the histology showing the highest frequency of *ERBB2* amplification, up to 38%, followed by other subtypes likewise clear cell or endometrioid carcinoma as pointed out by a study evaluating 2042 endometrial carcinomas [43]. These kinds of cases were found to be high-grade and associated with *TP53* alterations or CNV of *MDM2* which is concordant with our two cases that expressed *ERBB2* amplification concomitantly to *TP53* or *MDM2*. In fact, current guidelines recommend *ERBB2* testing in advance stage or recurrent serous endometrial cancer [44]. Despite of the low prevalence of *ERBB2* alterations in gynecological tumors, the good responses observed to these therapies warrant the need to determine this genomic aberration. In addition, a few representations of altered genes *FBXW7*, *PPP2R1A,* and *FGFR2* were encountered in the serous histological subtype. Recent studies demonstrate the relevance of *PPP2R1A* somatic alterations that may contribute to poor prognosis in patients with advance stage endometrial cancer independent of the histological subtype [45]. On the other hand, the endometrial endometrioid carcinoma was mainly characterized by somatic mutations in the *PTEN* gene (71%) which is the most common genomic aberration in this concrete subtype, occurring between 63% and 82% of endometrioid affected cases [14,46]. The *PIK3CA* gene was situated as the second more altered gene (60%), which is in accordance with described molecular genetics data in endometrial carcinomas [47]. An unexpected finding concerned the SNV of the *NOTCH1* gene occurring in 48% of the endometrial endometrioid cases. Precisely, each detected variant for this gene was categorized as Tier III or variant of uncertain significance. *NOTCH1* variants were called in a retrospective study conducted in 299 gynecological cancers identifying somatic mutations in the clear cell carcinoma subtype [48]. Other altered genes labelling the endometrioid cancer were *CTNNB1*, *KRAS*, *POLE,* and *ARID1A* genes, each of them accepted as genetic biomarker of the endometrioid tumors [49]. We found two cases of low-grade endometrioid carcinoma with altered *CTNNB1* that is thought to show prognostic significance with worsen recurrence-free survival in early-stage low-grade endometrial carcinoma. It has recently been reported that this specific subtype with altered *CTNNB1* could be incorporated in the future as the fifth histomolecular entity [39]. Finally, cervical HPV-associated cases, particularly those exhibiting a squamous cell carcinoma subtype were more altered for the *PIK3CA* gene (60%). Indeed, the *PIK3CA* gene is described as the top altered gene in cervical carcinoma along with the *MTOR*, *KMT2D* and *FAT1* genes [50]. Overall, such a molecular characterization seems to define the distinct studied gynecological cancers.

Although we conducted the study in a modest cohort of 49 patients, we compared only frequencies of single somatic mutations affected cases with those listed in the TCGA database. Only *PIK3CA* from uterus and cervix were reasonably close whereas *TP53* and *PTEN* were somehow distant from the TCGA frequencies.

Despite of obtaining a 2% Tier I variants in the *BRCA1* and *BRCA2* genes, as little as 4% of the patients received a PARPi treatment, however another 10% received PARPi exhibiting Tier III alteration in genes of the HRR pathway that summed up to a 14% of the cases. Such a scarce number of Tier I alterations found in our study raise some uncertainty about the utility of a comprehensive panel in the clinical practice of gynecological cancer. In fact, some studies have reported that 15% of patients undergoing comprehensive genotype profiling received genotype-guided therapies in solid tumors [51]. Alterations in the *POLE* gene are associated with hypermutated tumors and improved prognosis [52,53]. Most of the *POLE* mutated patients were classified as Tier III except one case interpreted as Tier II. A study analyzing 453 advanced tumors with aberrant *POLE* mutations also described an elevated number of variants of uncertain significance, about 69% of the studied cohort presented *POLE* variants of uncertain significance [54]. Despite of missing medical records regarding the radiological response it remains essential to consider a comprehensive sequencing panel including the analysis of the *POLE* gene. Considering the endometrial endometrioid case was at an initial stage no recurrence was seen, which is a good prognostic marker. On the other hand, the *PIK3CA* mutation in SSC, apart from its utility in molecularly defining the precise tumor type, could also be used as a molecular target as has been seen in another solid tumor such as in breast cancer and its associated guided-therapy alpelisib plus fulvestrant [55].

While immune checkpoint inhibitors (ICIs) have been successfully incorporated into the main therapeutic guidelines as the first line of treatment in some solid tumors such as in lung carcinoma and melanoma [56], they are not currently considered front-line in any gynecologic carcinoma types.

One of the main difficulties faced by ICIs is that there is not an established gold standard biomarker to select the most responsive patients to this kind of treatments [57]. In our study, we have analyzed two of the most used biomarkers to choose patients who are best candidates for ICIs [58], the percentage of TILs and the protein expression of PD-L1.

In endometrial cancer, we have observed an association between these two biomarkers, with higher levels, simultaneously, in the carcinomas considered to be of greater “immunogenicity”, in the dMMR-hypermutated and in the *POLE*-ultramutated. This has already been reported in diverse studies [58,59,60] and suggest that TILs, PD-L1 [61], and tumor mutational burden (TMB) [62] could be excellent alternatives as biomarkers, especially in the *POLE*-ultramutated subtype, since determining the POLE mutational status may require more complex and expensive molecular techniques. In contrast, in dMMR carcinomas, a subtype which already has an FDA-approved ICIs in advanced dMMR/MSI-H carcinomas [57], the mismatch repair immunohistochemistry is an acceptable assay to select these carcinomas [56]. However, further investigation would be required to elucidate whether the response rates are higher in dMMR/MSI-H tumors with higher levels of PD-L1 and/or TILs.

In endometrial cancer, certain genetic signatures are investigated as alternative biomarkers for ICIs, such as *ARID1A* alterations, associated with higher TILs, supporting the view that deficient *ARID1A* might be a potential predictor factor for ICIs efficacy [59]. On the contrary, in non-inmunogenic endometrial cancer, recent studies have associated alterations such as *CTNNB1* or *PIK3CA* genomic aberrations and *CMYC* amplification, with a low neoantigen load, that could lead to predict poor effects of immunotherapy response. So far, they could be used as potential candidates for ICIs. In our study, those endometrial cancers with an *ARID1A* mutation showed elevated levels of TILs and PD-L1 scores, and in contrast, cases with *CTNNB1* mutated showed low levels of these two biomarkers, which would support this hypothesis.

On the other hand, we have also seen high levels of these two biomarkers in cervical squamous carcinoma, also reflected in the literature [63]. At present, among gynecological cancer, only in advanced cervical carcinomas there is an FDA-approved ICI in which PD-L1 immunostaining is enlisted for checkpoint inhibitor access based on PD-L1 CPS status ≥1 as stated in the phase 2 Keynote 158 [64] and phase 3 Keynote 826 trials [65].

Among the ovarian carcinomas from our study, we observed the highest levels of PD-L1 and TILs in HGSC which is consistent with data reported in other studies [66,67,68]. The treatment of this subtype is based on PARPi, and it has been suggested that these therapies induce the formation of neoantigens generating an immune response. This would support this subtype as a good candidate to receive ICI in combination with PARPi and in consequence determining TILs and PD-L1 would be good candidates for patient eligibility [68]. Although in lower percentages than in endometrial cancer, in ovarian cancer there are also cases exhibiting a *POLE* mutation and dMMR status [66,68], thereby the presence of these alterations could determine a niche to offer ICIs in ovarian cancer.

Although all these contributions need further research in gynecological tumors, it is unreasonable to think of a scenario contemplating an analysis of several ICIs biomarkers performed simultaneously instead of one of them at once. That would be to consider the analysis at the time of diagnosis of TILs and PD-L1 scores as well as the MMR, *POLE,* and *ARID1A* mutational status aimed to determine precisely which patients could be more appropriate to respond to these treatments. In our study, we demonstrate the feasibility to perform this kind of study in the clinical practice. In fact, there are many clinical trials underway in gynecological cancers and the approval of checkpoint inhibitors in combination with other treatments should be forthcoming [69]. The selection of patients more sensitive to immunotherapy remains vital and for this reason the study of immune markers concomitant to molecular biomarkers urge to be studied in patients affected by gynecological cancer.

Several limitations may be recognized in this study. The compilation of data in 49 patients probably explained the limited description of additional tumor types in gynecological cancer. The illustration of further subtype such as ovarian low-grade serous carcinoma or uterine carcinosarcoma, among others, is lacking from the performed analysis. In fact, the major number of cases appeared in HGSC as it is the most common subtype of the five principal ovarian subtypes [70]. Although the application of the comprehensive high-throughput panel permitted a confident molecular characterization of certain histology, some biomarkers are not present in the used panel such as *PTEN* CNV, TMB or large deletions. Commercially available tests for molecular profiling and widely spread through diagnostic laboratories include this kind of alterations that are lacking from our study [71]. As well, the limited sample size of uterus and cervix cases could probably have explained the dissimilarity found between the TCGA frequencies and those from our analysis regarding the *TP53* and *PTEN* genes. An additional limitation of our study concerned the heterogeneity of the selected cohort. Although we were able to characterize diverse histological subtypes by the analysis of both molecular and immune biomarkers, we could not point out whether genomic alterations originated as a result of chemotherapy treatment or disease progression.

## 5. Conclusions

Lastly, the new version of the 2020 WHO classification in female genital tumors has given a special value in key molecular events as well as integrated both morphological and molecular features to attain a more refined classification of gynecological tumors [72]. In this sense, we may conclude that a molecular characterization in the clinical setting would be able to define, at least, the distinct studied gynecological tumor types by using comprehensive high-throughput sequencing. Moreover, it also permitted the identification of relevant genomic alterations related to a specific genotype-guided therapy. Finally, a routine concomitant analysis of immune biomarkers may provide a better selection of candidates for immunotherapy.

## Figures and Tables

**Figure 1 diagnostics-12-01903-f001:**
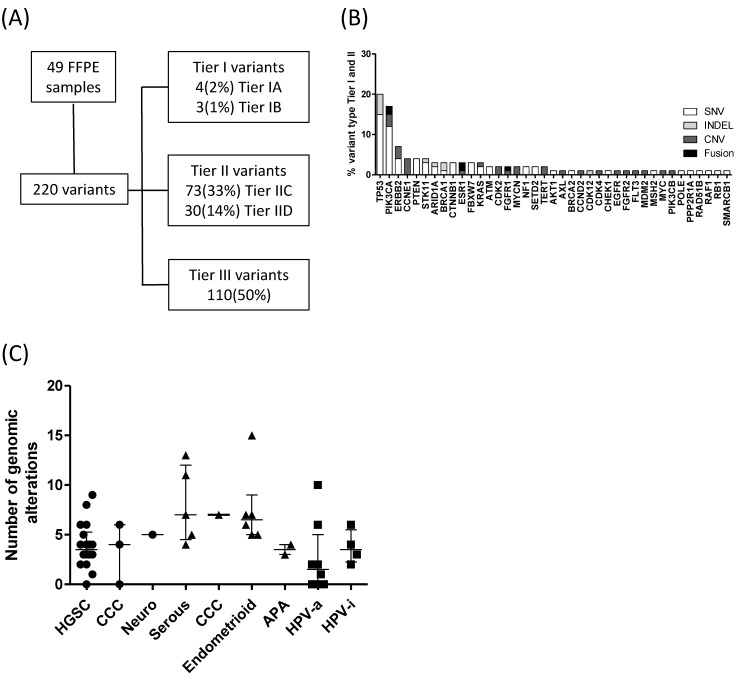
Classification and assessment of detected alterations. (**A**). Variant categorization based on AMP, ASCO, CAP guidelines, 2017. (**B**). Tier I and II variant types in the entire cohort by gene. (**C**). Number of genomic alterations by histological subtype. Ovarian and uterine samples are represented by circles and triangles, respectively, whereas cervical or vaginal samples are shown by squares. APA = atypical polypoid adenomyoma, CCC = clear cell carcinoma, HGSC = high-grade serous carcinoma, HPV-i = human papillomavirus-independent, HPV-a = HPV-associated, Neuro = neuroendocrine carcinoma.

**Figure 2 diagnostics-12-01903-f002:**
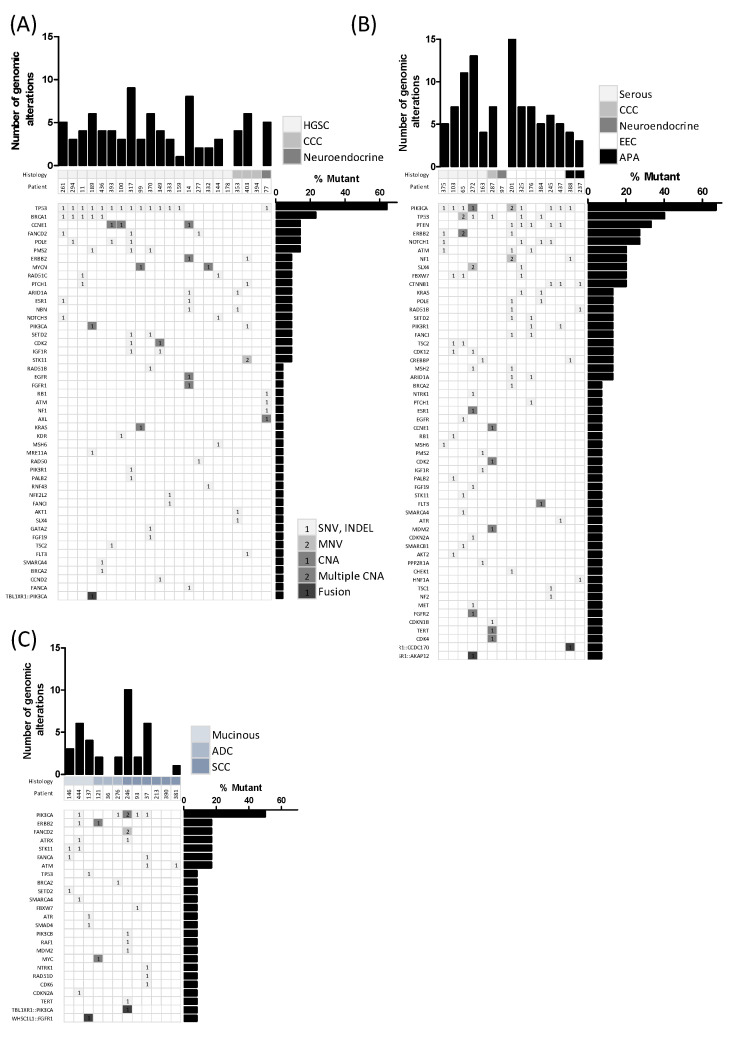
Detected genomic alterations across diverse histological subtypes. (**A**) Ovarian cancer, *n* = 22. (**B**) Uterine cancer, *n* = 15. (**C**) Cervical and vaginal cancer, *n* = 12. Columns represent samples and rows show genes expressed by the percentage of samples with a detected molecular alteration. Single nucleotide variant (SNV), small insertion and deletion (INDEL) and copy number alteration are shown by light grey, medium grey and dark grey squares, respectively, whereas fusions are depicted by black squares. Multiple nucleotide variant (MNV) and multiple CNA are shown by squares including inside more than one detected alteration in the same gene. ADC = adenocarcinoma, APA = atypical polypoid adenomyoma, CCC = clear cell carcinoma, EEC = endometrial endometrioid carcinoma, HGSC = high-grade serous carcinoma, SCC = squamous cell carcinoma.

**Figure 3 diagnostics-12-01903-f003:**
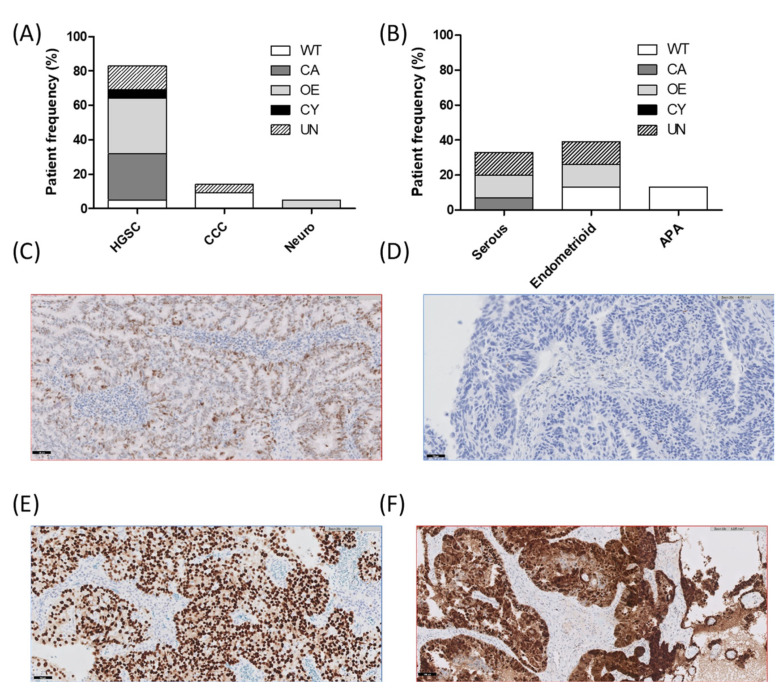
p53 protein expression. (**A**) Ovarian and (**B**) uterine frequencies of p53 immunohistochemical patterns stratified by histological subtype. (**C**) Representative image of a normal or wild-type (WT) pattern characterized by variable staining intensity. (**D**) Abnormal complete absence (CA) staining presents complete absence of expression in tumor nuclei. (**E**) Abnormal overexpression (OE) shows strongly intense staining in tumor nuclei. (**F**) Abnormal cytoplasmic expression (CY) shows diffuse cytoplasmic staining in the absence of strong nuclear staining. (Original magnification = 200× and scale bar = 50 microns). APA = atypical polypoid adenomyoma, CCC = clear cell carcinoma, HGSC = high-grade serous carcinoma, UN = unscored.

**Figure 4 diagnostics-12-01903-f004:**
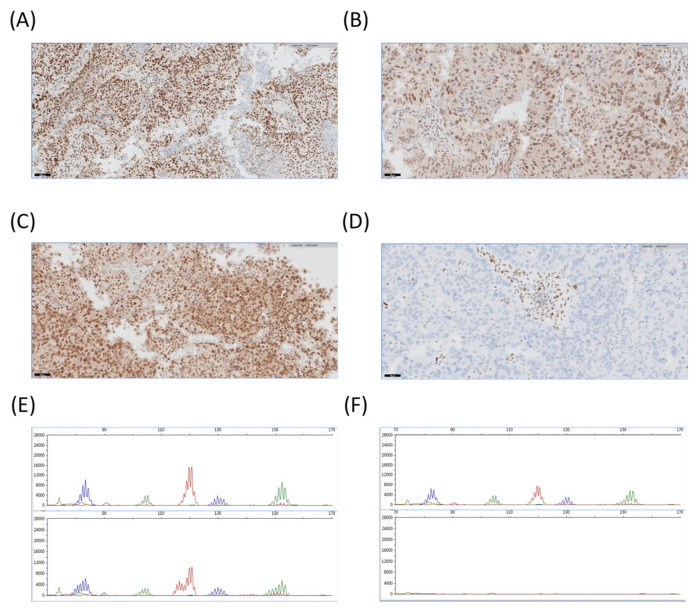
Deficient mis-matched repair (dMMR) protein expression and microsatellite instability. (**A**–**D**) Representative images of case 65, uterine serous carcinoma, exhibiting a concrete pattern of preserved protein expression of MLH1, PMS2, MSH2 and protein expression loss of MSH6. Original magnification = 200× and scale bars = 50 microns. (**E**) Representative electropherogram illustrating case 37, cervical squamous cell carcinoma, with a loss of stability in three out of five microsatellite markers, NR-27, BAT-25 and BAT-26. The upper graph shows normal tissue whereas the lower graph illustrates the tumor tissue. (**F**) Representative electropherogram showing a positive control of normal tissue (upper side) and a non-template control employed in the assay (lower side) defined by a non-amplification of the amplicons.

**Figure 5 diagnostics-12-01903-f005:**
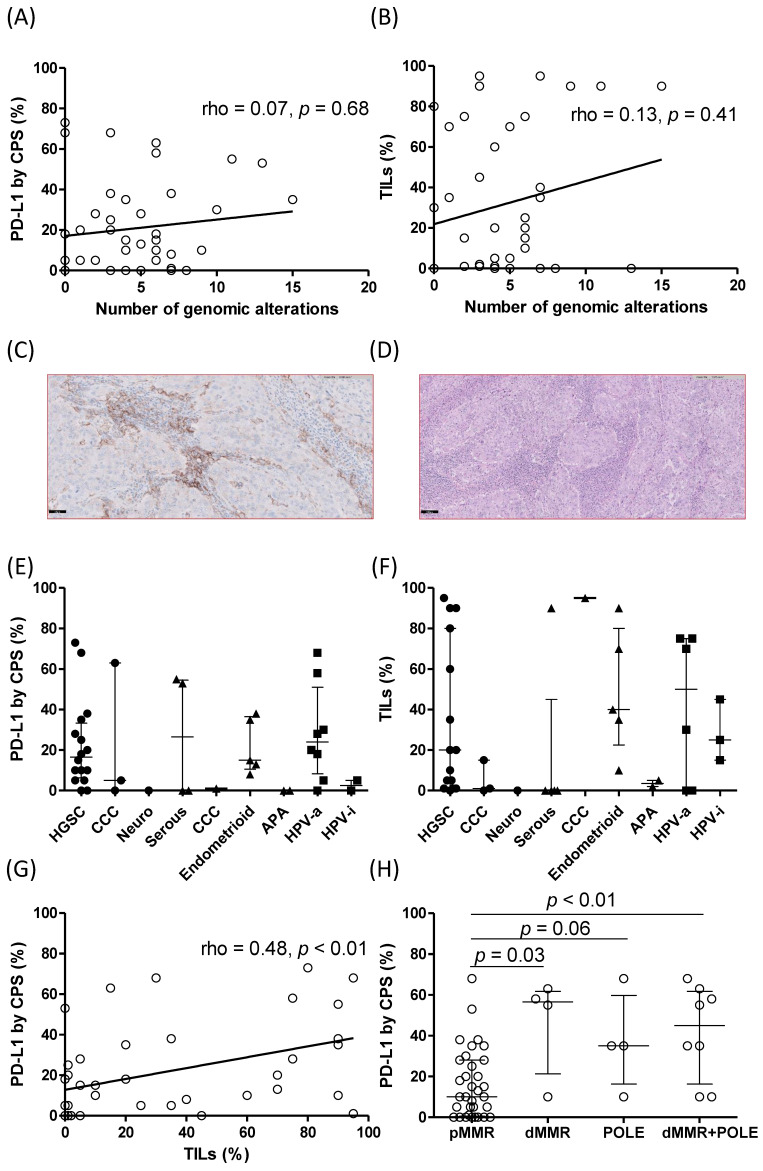
Evaluation of immune biomarkers (**A**) PD-L1 and (**B**) TiLs associated with genomic alterations. (**C**) Representative image of PD-L1 staining. Original magnification= 200×, scale bar = 50 microns (**D**) Representative image of a hematoxylin and eosin to score TILs. Original magnification =100×, scale bar = 100 microns (**E**) PD-L1 and (**F**) TILs stratified by histological subtype. (**G**) Association between PD-L1 and TILs in the entire cohort. (**H**) PD-L1 score by proficient or deficient mis-matched state and *POLE* mutation cases. Empty circles represent samples of the selected cohort. Ovarian and uterine samples are represented by black circles and triangles, respectively, whereas cervical or vaginal samples are shown by black squares. APA = atypical polypoid adenomyoma, CCC = clear cell carcinoma, HGSC = high-grade serous carcinoma, HPV-i = human papillomavirus-independent, HPV-a = HPV-associated, Neuro = neuroendocrine carcinoma.

**Table 1 diagnostics-12-01903-t001:** Clinicopathological characteristics of cohort, *n* = 49.

Feature, *n* (%)	Tumor Site			
	Ovary, *n* = 22	Uterus, *n* = 15	Cervix, *n* = 11	Vagina, *n* = 1
Median age (range)	56 (46–65)	61 (37–64)	53 (36–58)	46
Family history of malignancy	9 (18)	3 (6)	1 (2)	0
Metastatic disease	2 (4)	2 (4)	4 (8)	0
Histological diagnosis				
High grade serous carcinoma	18 (37)	0	0	0
Endometrioid carcinoma	0	6 (12)	0	0
Serous carcinoma	0	5 (10)	0	0
Atypical polypoid adenomyoma	0	2 (4)	0	0
Clear cell carcinoma	3 (6)	1 (2)	0	0
Neuroendocrine carcinoma	1 (2)	1 (2)	0	0
Squamous cell carcinoma	0	0	5 (10)	1 (2)
Mucinous carcinoma	0	0	3 (6)	0
Adenocarcinoma	0	0	3 (6)	0
FIGO stage *				
I	1 (2)	5 (10)	1 (2)	1 (2)
II	0	1 (2)	0	0
III	15 (31)	2 (4)	1 (2)	0
IV	5 (10)	4 (8)	8 (16)	0
Unknown	1 (2)	3 (6)	1 (2)	0
Treatment scheme				
Neo-adjuvant	12 (24)	2 (4)	6 (12)	0
Adjuvant	9 (18)	5 (10)	3 (6)	1 (2)
None	0	7 (14)	0	0
Unknown	1(2)	1 (2)	2 (4)	0
Treatment regimen				
Platinum	0	0	4 (8)	1 (2)
Platinum+taxane	19 (39)	7 (14)	3 (6)	0
Platinum+taxane+bevacizumab	1 (2)	0	2 (4)	0
Others	1 (2)	1 (2)	2 (4)	0
None	0	6 (12)	0	0
Unknown	1 (2)	1 (2)	0	0
Adjuvant radiotherapy	1 (2)	6 (12)	10 (20)	0
Surgery debulking	19 (39)	13 (26)	7 (14)	0
Recurrence	17 (35)	9 (18)	7 (14)	1 (2)
Radiological response				
Complete response	6 (12)	0	1 (2)	0
Partial response	15 (31)	5 (10)	6 (12)	1 (2)
Progressive disease	0	2 (4)	3 (6)	0
Unknown	1 (2)	8 (16)	1 (2)	0

* FIGO = International Federation of Gynecology and Obstetrics.

**Table 2 diagnostics-12-01903-t002:** Molecular characterization of gynecological histological subtypes.

Molecular Pathway	Tumor Site (Histological Subtype)							
	Ovary		Uterus			Cervix		
	H-G Serous Carcinoma	Clear Cell Carcinoma	Serous Carcinoma	Clear Cell Carcinoma	Endometrioid Carcinoma	Squamous Cell Carcinoma	Adenocarcinoma	Mucinous Carcinoma
HPV	-	-	-	-	-	associated		independent
TP53	TP53 mut	-	TP53 mut	TP53 mut	-	-	-	-
Wnt-beta-catenin	-	-	CTNNB1 mut	-	CTNNB1 mut	-	-	-
			SOX17 CNA					
PiK3CA-PTEN-AKT-mTOR	PiK3CA CNA	PiK3CA mut	PiK3CA mut+CNA	PiK3CA mut	PiK3CA mut	PiK3CA mut	PiK3CA mut	-
		PTEN loss	PTEN mut	PTEN loss	PTEN mut			
			PiK3R1 mut		PiK3R1 mut			
MAP kinase	NF1 mut	KRAS mut	-	-	KRAS mut	-	KRAS mut	-
Tyrosine kinase receptors	-	-	ERBB2 CNA	ERBB2 mut+CNA		ERBB3 mut	-	ERBB2 CNA
			FGFR2 mut		FGFR2 mut			
			FGFR1 CNA					
			FGFR3 CNA					
Homologous recombination deficiency	BRCA1 mut	-	-	-	-	-	-	-
	BRCA2 mut							
	CDK12 mut							
	EMSY mut							
	BRIP mut							
	PALB2 mut							
	RAD51 mut							
	BARD1 mut							
	ATM mut							
	ATR mut							
Mismatch repair	-	-	-	MSI-H	MSI-H	-	-	-
Base excision repair	-	-	-	-	POLE mut	-	-	-
SWI/SNF nucleosome remodeling complex	-	ARID1A mut	-	ARID1A mut	ARID1A mut	-	-	-
Cell cycle	CCNE1 CNA	-	CCNE1 CNA	-	-	-	-	MDM2 CNA
	RB1 mut+CNA		MYC mut					
	CDKN2A mut		PPP2R1A mut					
Other genomic aberrations		TERT mut		TERT mut	-	CASP8, HLA-A, SHKBP1, TGBR2, TGFbeta	-	
	FOXM1 CNA		FBXW7 mut					STK11 mut
	NOTCH1 CNA		LRPB1 loss					

ARID1A = AT-Rich Interaction Domain 1A 2, ATM = ATM Serine/Threonine Kinase, ATR = ATR Serine/Threonine Kinase, BARD1 = BRCA1 Associated RING Domain 1, BRCA = BRCA1 DNA Repair Associated, BRIP = BRCA1 Interacting Protein C-Terminal Helicase 1, CASP8 = Caspase 8, CCNE1 = Cyclin E1, CDK12 = Cyclin Dependent Kinase 12, CDKN2A = Cyclin Dependent Kinase Inhibitor 2A, CNA = copy number amplification, CTNNB1 = Catenin Beta 1, dMMR = deficient Mis-Match Repair, EMSY = EMSY Transcriptional Repressor (BRCA2 Interacting), ERBB2(3) = Erb-B2 Receptor Tyrosine Kinase 2(3), FBXW7 = F-Box And WD Repeat Domain Containing 7, FGFR2(1,3) = Fibroblast Growth Factor Receptor 2(1,3), FOXM1 = Forkhead Box M1,HLA-A = Major Histocompatibility Complex (Class IA), KRAS = KRAS Proto-Oncogene (GTPase), LRPB1 = Low-Density Lipoprotein Receptor Related Protein 1B, MDM2 = Proto-Oncogene (E3 Ubiquitin Protein Ligase), MSI-H = Microsatellite instability-high, mut = mutation, MYC = MYC Proto-Oncogene, BHLH Transcription Factor, NF1 = Neurofibromin 1, Notch1 = Notch Receptor 1, PALB2 = Partner And Localizer Of BRCA2, PiK3CA = Phosphatidylinositol-4,5-Bisphosphate 3-Kinase Catalytic Subunit Alpha, PiK3R1 = Phosphoinositide-3-Kinase Regulatory Subunit 1, POLE = DNA Polymerase Epsilon (Catalytic Subunit), PPP2R1A = Protein Phosphatase 2 Scaffold Subunit Alpha, PTEN = Phosphatase And Tensin Homolog, RAD51 = RAD51 Recombinase, RB1 = RB Transcriptional Corepressor 1vRB Transcriptional Corepressor 1, SHKBP1 = SH3KBP1 Binding Protein 1, SOX17 = SRY-Box Transcription Factor 17, STK11 = Serine/Threonine Kinase 11, TERT = Telomerase Reverse Transcriptase, TGFbeta = Transforming Growth Factor Beta, TGBR2 = Transforming Growth Factor Beta Receptor 2, TP53 = Tumor protein p53.

**Table 3 diagnostics-12-01903-t003:** Relationship between immune markers and genomic alterations.

Patient	Tumor Site	Histological Subtype	MMR Status	Microsatellite Instability	Repair Altered Genes	PD-L1 by CPS (%)	TiLs (%)
384	Uterus	Endometrioid	-	MSS	POLE	-	-
375	Uterus	Serous	pMMR	MSS	MSH6	-	0
163	Uterus	Serous	pMMR	MSS	PMS2	0	0
317	Ovary	HGSC	pMMR	MSS	PMS2/POLE	10	90
189	Ovary	HGSC	pMMR	MSI-L	PMS2	10	10
370	Ovary	HGSC	pMMR	MSS	PMS2	18	20
144	Ovary	HGSC	pMMR	MSS	MSH6	20	1
393	Ovary	HGSC	pMMR	-	POLE	35	20
201	Uterus	Endometrioid	pMMR	MSS	POLE	35	90
272	Uterus	Serous	pMMR	MSS	MSH2	53	0
65	Uterus	Serous	dMMR	MSI-L	NONE	55	90
37	Cervix	SCC	dMMR	MSI-L	NONE	58	75
403	Ovary	CCC	dMMR	MSI-H	NONE	63	15
294	Ovary	HGSC	-	MSS	POLE	68	95

CPS = combined positive score, HGSC = high-grade serous carcinoma, SCC = squamous cell carcinoma, CCC = clear cell carcinoma, dMMR = deficient mis-matched repair, pMMR = proficient mis-matched repair, MSS = microsatellite stability, MSI = microsatellite instability, MSI-L = MSI-low, MSI-H = MSI-high.

**Table 4 diagnostics-12-01903-t004:** Utility of clinically relevant alterations and concrete targeted therapy.

Patient	Gene	Variant Categorization	Variant Type	Tumor Site	Histological Subtype	Targeted Therapy	Radiological Response
11	BRCA1	Tier IA	INDEL	Ovary	HGSC	None	CR
163	PPP2R1A	Tier IID	SNV	Uterus	Serous	Olaparib	PR
178	WT	-	-	Ovary	HGSC	Niraparib	CR
189	BRCA1	Tier III	SNV	Ovary	HGSC	None	PR
201	BRCA2	Tier IA	SNV	Uterus	Endometrioid	None	Lost to follow-up
261	BRCA1	Tier III	SNV	Ovary	HGSC	Olaparib	PR
276	BRCA2	Tier III	SNV	Cervix	Endocervical	None	CR
294	BRCA1	Tier IA	SNV	Ovary	HGSC	Olaparib	PR
370	RAD51B	Tier III	SNV	Ovary	HGSC	Niraparib	CR
375	ATM	Tier III	SNV	Uterus	Serous	Olaparib	Lost to follow-up
381	ATM	Tier III	SNV	Vagina	SSC	Pembrolizumab *	PR
390	WT	-	-	Cervix	SSC	Pembrolizumab *	PR
436	BRCA1-BRCA2	Tier IA- Tier III	INDEL-SNV	Ovary	HGSC	Olaparib	PR

* Immune biomarker PD-L1 resulted positive at the time of diagnosis.

## Data Availability

Datasets used in the current study may be available on a reasonable request.

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
