# Peer review of "Comprehensive Approach to Genomic and Immune Profiling: Insights of a Real-World Experience in Gynecological Tumors"

_diagnostics, 2022, doi:10.3390/diagnostics12081903_

Round 1
Reviewer 1 Report
Prieto-Potin I et al., reported on a comprehensive approach to genomic and immune profiling: insights of a real-world experience in gynecological tumors. The authors performed a characterization of FFPE tissues from gynecological tumors undergoing high-throughput sequencing. The sequencing panel consists of 161 cancer-associated genes. The study aims to determine whether the application of high-throughput sequencing established as part of the clinical care practice can characterize a particular histological subtype of gynecological tumors by the detection of precise alterations. Despite the manuscript presented including an interesting approach, it appears limited in a way with the sequencing of only 161 genes. By applying high-throughput sequencing, did the authors identify any novel genomic aberration associated with a specific histotype? Additional concerns also include the heterogeneity of the cohort. The cohort includes adjuvant and neoadjuvant treated samples in different stages of the disease (debulking surgery primary diagnosis or recurrence), how the authors can be sure that some of the mutations identified are not induced by the chemotherapeutic treatment or during disease progression?
Minor points
Figure 2 is very complex and difficult to read the authors should present choose a clearer presentation of the results.
In figure 3 the correlation values are missing.
Representative images of the staining performed throughout the manuscript should be presented.
The sentence “The Among gynecological cancer, three types account for the vast majority of new” should be revised
Author Response
Comments and Suggestions for Authors
Prieto-Potin I et al., reported on a comprehensive approach to genomic and immune profiling: insights of a real-world experience in gynecological tumors. The authors performed a characterization of FFPE tissues from gynecological tumors undergoing high-throughput sequencing. The sequencing panel consists of 161 cancer-associated genes. The study aims to determine whether the application of high-throughput sequencing established as part of the clinical care practice can characterize a particular histological subtype of gynecological tumors by the detection of precise alterations. Despite the manuscript presented including an interesting approach, it appears limited in a way with the sequencing of only 161 genes. By applying high-throughput sequencing, did the authors identify any novel genomic aberration associated with a specific histotype? Additional concerns also include the heterogeneity of the cohort. The cohort includes adjuvant and neoadjuvant treated samples in different stages of the disease (debulking surgery primary diagnosis or recurrence), how the authors can be sure that some of the mutations identified are not induced by the chemotherapeutic treatment or during disease progression?
Response: Thank you very much for your critical comments.
The high-throughput panel used in the present study identifies 161 genes. In fact, most of the genes molecularly describing the histological subtypes, as shown in Table 2, are included in the employed panel (please see Table S1). We have stated in the manuscript those few genes and alterations types that are missing, thus being part of the limitations of the conducted research, for instance, loss of PTEN or identification of large deletions. This statement was cited in the discussion. According to Hynes and co-workers, the design of gene panels in next-generation sequencing can be classified into 4 groups [1]. The panel used in our study may belong to a class 3 design or tumor comprehensive panel, up to 150 genes, which is along with a class 2 design the most widely used panels in diagnostics and clinical trials. Therefore, we contemplated the OCA v3 panel as suitable for the performed investigation.
We did not identify novel genomic aberrations. We have only detected NOTCH1 SNV in uterine endometrioid carcinoma. However, any conclusion could not be gained because of the limited sample size of the specific subtype that does not permit to rule out a plausible association to a concrete histology. Most relevant detected alterations in our study were broadly described in the literature and concordant with the different analyzed subtypes.
We agree the reviewer’s opinion regarding the heterogeneity of the studied cohort. The selection of cases was done according to those cases who were asked for high-throughput sequencing during routine practice and not based in a research criterion. In any case, we cannot be sure if some of the mutations may be induced by chemotherapy or during disease progression as the study design was not defined to trigger this kind of data. For this purpose, a different cohort discriminating neoadjuvant and adjuvant treated paired samples might be chosen in order to recognize associations with chemotherapeutic therapies or progression. The obtained findings are core or previous to treatment.
Action: We have included in the discussion section the following statement: “An additional limitation of our study concerned the heterogeneity of the selected cohort. Although we were able to characterize by the analysis of both molecular and immune biomarkers diverse histological subtypes, we could not trigger whether genomic alterations were originated by chemotherapy treatment or disease progression.”
Minor points
Figure 2 is very complex and difficult to read the authors should present choose a clearer presentation of the results.
Response: Thank you very much for your suggestion. Indeed, figure 2 was complex and difficult to read.
Action: We have now split Figure 2 into several figures in order to present the results in a clearer fashion. We left Figure 2 along with the detected high-throughput alterations stratified by histological subtypes. Then we have created Figure 3 and Figure 4 to show p53 protein expression and deficient MMR/MSI results. Additional Figures S1 and S2 display representative images of ERBB2 protein expression and confirmation by FISH, as well as HPV genotyping. Finally, Figure 3 was transformed into Figure 5 by including both representative images of PD-L1 and TILs.
In figure 3 the correlation values are missing.
Response: Thank you for the observation.
Action: We have added correlation values in figure 3 where appropriated.
Representative images of the staining performed throughout the manuscript should be presented.
Response: Thank you for your recommendation.
Action: We have included representative images of the performed staining throughout the entire manuscript. IHC staining of p53, MMR, ERBB2, PD-L1 and hematoxylin and eosin for assessing TILs have been added.
The sentence “The Among gynecological cancer, three types account for the vast majority of new” should be revised
Response: Thank you for this observation.
Action: We have revised the entire manuscript in order to correct typo-mistakes including the above mentioned regarding the first sentence of the introduction.

Reviewer 2 Report
(1) Please evaluate the cost-effectiveness of this profiling protocol.
(2) Except for cervical cancer, immunotherapy has not yet been proven as an effective treatment for other gynecologic cancers. Is it appropriate to use immunotherapy based on the results of immune marker analysis for these patients?
(3) What clinical benefit could the ovarian cancer patients get if they were known to have amplifications in the CCNE1 gene or TP53 mutation?
(4) Atypical polypoid adenomyoma is not a frankly invasive disease. Did the two cases of uterine atypical polypoid adenomyoma also undergo subsequent genomic and immune profiling?
(5) Please mention the variant categorization (Tier IA, IB, IIC, IID, and III) in the Materials & Methods section.
(6) In table 1, “(2018 FIGO)” can be omitted. Otherwise, modify it to “(2018 FIGO for cervical cancer)”.
Author Response
- Please evaluate the cost-effectiveness of this profiling protocol.
Response: Thank you for this proposal. We did not collect data about costs and survival from the studied patients as well as related complications and events derived from their disease and administered treatments. For this reason, we are unable to perform a proper cost-effectiveness assessment of our profiling protocol. As an alternative, we suggest to evaluate the costs of single gene test or routine conventional testing in comparison to the employed NGS panel in the histological subtype with a major representation of the cohort.
Action: We have included a brief comment stating the costs of the presented profiling protocol in the results section and a supplementary table (Table S6) describing the costs of either single gene tests or conventional testing in comparison to the OCA v3 NGS, as well as the immune profiling:
“3.6 Total cost and cost difference versus NGS
We performed an evaluation of the conventional testing strategy versus the high-throughput panel used for those more clinically relevant biomarkers in high-grade serous ovarian carcinoma. The cost per sample, the testing strategy, total cost and cost difference is depicted in Table S6. The application of the profiling protocol including the comprehensive NGS panel and the identification of immune biomarkers (PD-L1 and TILs) resulted more profitable than the use of conventional or single gene testing.”
Table S6. Testing strategy, total cost and cost difference versus OCA NGS
|
n=1 |
Ovarian HGSC patients, n=22 |
|
Testing strategy |
Testing cost |
Total cost |
Cost difference versus OCA NGS |
TP53 |
150 |
3300 |
6512 |
BRCA NGS |
404 |
8888 |
924 |
PALB2 |
150 |
3300 |
6512 |
RAD51 |
150 |
3300 |
6512 |
ERBB2 |
180 |
3960 |
5852 |
Total conventional testing |
1034 |
22748 |
12936 |
OCA NGS |
446 |
9812 |
- |
PD-L1 |
161 |
3542 |
- |
TILs |
4 |
88 |
- |
Total OCA NGS and immune testing |
611 |
13442 |
3630 |
Costs are given in 2017 euros, HGSC=high-grade serous carcinoma, NGS=next-generation sequencing, OCA=oncomine comprehensive assay
(2) Except for cervical cancer, immunotherapy has not yet been proven as an effective treatment for other gynecologic cancers. Is it appropriate to use immunotherapy based on the results of immune marker analysis for these patients?
Response: Thank you for this critical comment.
Certainly, at present, only in advanced cervical carcinoma there is an FDA-approved immune checkpoint inhibitor based on the PD-L1 status and based in the same PD-L1 criterion it has not been proven to be effective in other gynecological cancers yet.
It does not seem by now that immunotherapy based on immune marker analysis is appropriate for these patients. In fact, immunotherapy may be indicated in those cases exhibiting a MMRd/MSI-H status and high TMB in endometrial cancer and less frequently in ovarian cancer [1,2].
However, it is not unreasonable to consider that such criterion based on PD-L1 or TILs analysis could be extended to other neoplasia in a near future, for instance, vulvar and vagina carcinomas, since they are biologically overlapping with cervical carcinoma and probably to either endometrial or ovarian cancer, especially in POLE mutated carcinomas that might present high levels of TILs and PD-L1.
In our study, we have shown the feasibility to evaluate both immune biomarkers, TILs and PD-L1. As new combinations with other drugs are under investigation [3], we speculate that the study of these biomarkers could result useful for the clinical setting.
(3) What clinical benefit could the ovarian cancer patients get if they were known to have amplifications in the CCNE1 gene or TP53 mutation?
Response: Thank you for this remark.
CCNE1 amplification is associated with poorer responses to first-line platinum chemotherapy and is also associated to a mechanism of immune resistance [4]. There is not a direct clinical benefit but rather it is considered a predictive biomarker. In addition, CCNE1 amplification in HGSOC is mutually exclusive of homologous recombination pathway mutations, including BRCA alterations, resulting in an ineffective PARP inhibition [5,6]
The TP53 mutation in ovarian cancer may aid to classify a particular histological subtype, precisely high-grade serous carcinoma. Despite of the diverse controversial data regarding the precise role of p53 in the progression and therapy response in ovarian carcinoma, p53 has indeed demonstrated utility as a diagnostic biomarker. For instance, if a concomitant BRCA mutation is detected an indirect clinical benefit may be obtained by the administration of inhibitors of PARP.
Action: We have introduced a brief statement regarding CCNE1 in the discussion section: “CCNE1 amplification in HGSOC is mutually exclusive of homologous recombination pathway mutations, including BRCA alterations, resulting in an ineffective PARP inhibition.”
(4) Atypical polypoid adenomyoma is not a frankly invasive disease. Did the two cases of uterine atypical polypoid adenomyoma also undergo subsequent genomic and immune profiling?
Response: Thank you. We completely agree the reviewer’s comment.
Indeed, atypical polypoid adenomyoma (APA) is an endometrial benign lesion not frankly invasive that tends to recur frequently. It is associated with or precedes endometrial atypical hyperplasia or endometrioid adenocarcinoma [7,8]. A close follow-up is recommended in this kind of uncommon uterine lesions.
Both APA cases included in our study correspond to young women who have not develop any adenocarcinoma to date. Indeed, both cases underwent genomic profiling but not the analysis of immune biomarkers. PD-L1 and TILs studies were performed retrospectively once the cases were selected for the study.
(5) Please mention the variant categorization (Tier IA, IB, IIC, IID, and III) in the Materials & Methods section.
Response: Thank you very much for your suggestion.
Action: We have mentioned in the M&M section the variant categorization: “Variant categorization was assessed as previously described [23,24]. Briefly, Tier IA and IB categories corresponded to variants with strong clinical significance, Tier IIC and IID matched to variants with potential clinical significance and Tier III agreed to variants with unknown clinical significance.”
(6) In table 1, “(2018 FIGO)” can be omitted. Otherwise, modify it to “(2018 FIGO for cervical cancer)”.
Response: Thank you for your comment. We completely agree the reviewer’s opinion.
Action: we have omitted the statement 2018 FIGO in table 1.
